# MAPK Cascades and Transcriptional Factors: Regulation of Heavy Metal Tolerance in Plants

**DOI:** 10.3390/ijms23084463

**Published:** 2022-04-18

**Authors:** Shaocui Li, Xiaojiao Han, Zhuchou Lu, Wenmin Qiu, Miao Yu, Haiying Li, Zhengquan He, Renying Zhuo

**Affiliations:** 1State Key Laboratory of Tree Genetics and Breeding, Chinese Academy of Forestry, Beijing 100091, China; lishaocui@caf.ac.cn (S.L.); hanxiaojiao1004@163.com (X.H.); luzc@caf.ac.cn (Z.L.); qiuwm05@caf.ac.cn (W.Q.); myu@caf.ac.cn (M.Y.); 2Forestry Faculty, Nanjing Forestry University, Nanjing 210037, China; 3Key Laboratory of Tree Breeding of Zhejiang Province, The Research Institute of Subtropical Forestry, Chinese Academy of Forestry, Hangzhou 311400, China; 4Institute of Virology and Biotechnology, Zhejiang Academy of Agricultural Sciences, Hangzhou 310021, China; lihaiying_2004@163.com; 5Key Laboratory of Three Gorges Regional Plant Genetic and Germplasm Enhancement (CTGU), Biotechnology Research Center, China Three Gorges University, Yichang 443002, China

**Keywords:** MAPK, transcriptional factors, heavy metals, pathway

## Abstract

In nature, heavy metal (HM) stress is one of the most destructive abiotic stresses for plants. Heavy metals produce toxicity by targeting key molecules and important processes in plant cells. The mitogen-activated protein kinase (MAPK) cascade transfers the signals perceived by cell membrane surface receptors to cells through phosphorylation and dephosphorylation and targets various effector proteins or transcriptional factors so as to result in the stress response. Signal molecules such as plant hormones, reactive oxygen species (ROS), and nitric oxide (NO) can activate the MAPK cascade through differentially expressed genes, the activation of the antioxidant system and synergistic crosstalk between different signal molecules in order to regulate plant responses to HMs. Transcriptional factors, located downstream of MAPK, are key factors in regulating plant responses to heavy metals and improving plant heavy metal tolerance and accumulation. Thus, understanding how HMs activate the expression of the genes related to the MAPK cascade pathway and then phosphorylate those transcriptional factors may allow us to develop a regulation network to increase our knowledge of HMs tolerance and accumulation. This review highlighted MAPK pathway activation and responses under HMs and mainly focused on the specificity of MAPK activation mediated by ROS, NO and plant hormones. Here, we also described the signaling pathways and their interactions under heavy metal stresses. Moreover, the process of MAPK phosphorylation and the response of downstream transcriptional factors exhibited the importance of regulating targets. It was conducive to analyzing the molecular mechanisms underlying heavy metal accumulation and tolerance.

## 1. Introduction

Among a variety of soil pollutants, heavy metal pollution is a key worldwide environmental issue. Heavy metals (HMs) cannot be decomposed and will be available in the soil permanently. Heavy metals usually exist in the environment and interact with plants and human systems. The toxicity of HMs in the environment is the product of natural and human actions [1]. HMs are inorganic and non-biodegradable pollutants which are not easy to metabolize. Therefore, their concentrations in soil are increasing significantly [2]. In the surrounding environment, accidental biomagnification and bioaccumulation of heavy metals have become a predicament for all organisms, including plants [3]. 

A variety of heavy metals in the soil are absorbed by plants. These heavy metals may or may not be necessary for the normal growth of plants [4]. When the growth and development of plants are constantly stressed by heavy metals, their biological systems are irreversibly damaged, resulting in reductions in plant yield and productivity [5]. Therefore, plants respond to and adapt to these environmental challenges through a series of physiological and biochemical changes. This process involves a series of complex signal pathways [6,7]. Essentially, stress sources can induce intracellular signal perception through external signals (calcium or miRNA) [8]. On this basis, the signals sensed by cell membrane surface receptors are amplified step-by-step through phosphorylation and dephosphorylation and transmitted to cells [9]. In cells, signals can activate specific effector proteins, such as kinases, enzymes or transcriptional factors, in the cytoplasm or nucleus and regulate the expression of specific genes [10].

Heavy metal stress signals of different intermediate molecules activate different transcriptional factors, resulting in the expression of different antioxidant enzymes [11]. Among them, protein phosphorylation is an important signal transduction mode in plants, which is catalyzed by mitogen-activated protein kinases (MAPKs). MAPKs are just some of the most principal and highly conserved signaling molecules in eukaryotes. They function downstream of the sensor/receptor to coordinate cellular responses for normal growth and development of the organism [12]. MAPKs can be easily identified based on sequence similarity and the signature TXY activation motif and enable the transmission of signals generated by ligand–receptor interactions with downstream substrates [13].

There are various MAPK pathways in cells. Each pathway is interrelated and independent and plays a major role in cell signal transmission. In plants, the MAPK cascade is intertwined with other signal transduction pathways to form a molecular interaction network [14]. Diverse cellular functions in plants, including growth, development, biological and abiotic stress responses, are regulated by this network. For instance, MAPK in plants can target and regulate bZIP, MYB, MYC and WRKY transcriptional factors under stress conditions [15]. However, considerable progress has been achieved in understanding the mechanism of action of the MAPK cascade in plant innate immunity. Upon *Rhizoctonia cerealis* infection, strongly upregulated *TaMKK5* activates *TaMPK3*, and the phosphorylated *TaMPK3* interacts with and phosphorylates *TaERF3* [16]. However, it is seldom reported that the MAPK signal is induced and activated by downstream transcriptional factors under heavy metal stress. The roots are the plant’s main organ for uptake of heavy metal from the soil [17]. When the roots perceive heavy metal stress, they immediately trigger the signal transduction system and mediate the transcriptional regulation of related genes in the plants. Conserved MAPK signaling pathways are known to regulate cell growth and death, differentiation, the cell cycle and stress responses. MAPK is a series of phosphorylation steps from MAPKKKs (MKK kinases), via MAPKKs (MAPK kinases), to MAPK [18]. Its main function is to phosphorylate related transcriptional factors. Subsequently, transcriptional factors can induce metal response gene expression by binding specific cis-regulatory elements. Each hierarchy of the MAPK cascade is encoded by a small gene family, and multiple members can function redundantly in an MAPK cascade [19]. Plant MAPKs are usually located in the cytoplasm and/or nucleus, although they may also be transferred from the cytoplasm to the nucleus in some cases [20]. 

It is essential to understand the progress of signal transduction and regulation pathways in plants under heavy metal stress. Thus, we focused on the molecular mechanisms of heavy metals entering plant cells to produce reactive oxygen species, nitric oxide and plant hormones and activate the MAPK cascade signal pathway. In addition, the downstream transcriptional factors responding to genes of the MAPK cascade are also summarized (Figure 1). This review provides more comprehensive background knowledge of plants under heavy metal stress and new insights into the molecular mechanisms of transcriptional factors in heavy metal tolerance and accumulation.

## 2. MAPK Was Directly Activated under Heavy Metal Stress

As a cell signaling enzyme, MAPK regulates a variety of biological processes in eukaryotes [21]. MAPK pathways are very developed and complex and are usually induced to deal with biological and abiotic stress.

In plants, heavy metal stress initiates a variety of signal pathways, including the MAPK cascade (Table 1). In *Broussonetia papyrifera* roots, under cadmium (Cd) stress over time, MAPK transcripts were downregulated at 3 hours but upregulated at 6 h [22]. Moreover, *OsMPK3* and *OsMPK6* overexpression lines increased the transcription level of the stress response genes encoding superoxide dismutase, ascorbate peroxidase, glutamine synthase and aldehyde oxidase under arsenic stress and drought stress [23]. Furthermore, the *SlMAPK3* gene of the tomato was significantly induced under Cd^2+^ treatment. The overexpression of *SlMAPK3* significantly increased leaf chlorophyll content, root biomass accumulation and root activity in transgenic plants, demonstrating that *SlMAPK3* enhanced Cd tolerance [24].

## 3. Different Signal Molecules Activate MAPK Pathway under Heavy Metal Stresses

The MAPK cascade can interact with signal molecules such as plant hormones, active ROS and NO. The crosstalk between ABA, auxin, MAPK signaling and the cell cycle in Cd-stressed rice seedlings has also been described [38]. In *Arabidopsis thaliana,* exposure to excess Cd or copper (Cu) led to the activation of NADPH oxidases, hydrogen peroxide (H_2_O_2_) overproduction and MAPK cascades [39]. In addition, the roots of soybean seedlings treated with 25 mg·L^−1^ Cd showed increased NO production and the upregulation of the *MAKPK2* transcription level [40].

### 3.1. ROS

It is well known that reactive oxygen species in plants are induced by heavy metals; thereby, ROS, as signal molecules, lead to the activation of MAPK kinases [41]. Two important MAPK cascades (MEKK1-MKK4/5-MPK3/6 and MEKK1-MKK2-MPK4/6) act downstream of ROS, which were found to participate in both abiotic and biotic stresses [25,26]. As a redox active metal, a certain concentration of Cu^2+^ directly induces the formation of ROS. When *Alfalfa* seedlings were exposed to excessive Cu^2+^, ROS accumulated and activated four different mitogen-activated protein kinases (MAPKs): SIMK, MMK2, MMK3 and SAMK [27]. Except for Cu^2+^, Cd stress was able to activate *ZmMPK3-1* and *ZmMPK6-1* via ROS induction in maize roots [29]. Moreover, the activities of *MPK3* and *MPK6* increased significantly in Cd-treated *Arabidopsis* seedlings, whereas this increase disappeared in the plants pretreated with the ROS scavenger glutathione (GSH). The above results fully indicate that Cu^2+^- or Cd^2+^-induced ROS accumulation in plants activate MAPK cascade [28].

H_2_O_2_, as a product of oxidative stress, is involved in amplifying the functions of signal molecules. MAPK can also be activated by H_2_O_2_ to maintain intracellular homeostasis [42]. Furthermore, the overexpression of downstream MAPK may also be a signaling transmission mechanism after sensing H_2_O_2_ [43]. These signaling components contain at least three specific phosphorylated kinases (MAPK2, MAPK3 and MAPK6), which can be observed in all living cells. Moreover, excessive Cu led to the activation of NADPH and the excessive production of H_2_O_2_, thereby inducing the MAPK cascade in *Arabidopsis* roots [39]. All these results indicate that the MAPK cascade activated by ROS molecules can play an important role under different metal stresses.

### 3.2. NO 

NO is involved in plant growth and development and regulates heavy metal responses in plants [31]. In HM treated plants, the interaction between the NO signal and the MAPK cascade has long been known [32]. Application of NO to *Arabidopsis* roots can rapidly activate protein kinases with MAPK properties [30]. When two-week-old *Arabidopsis* was exposed to 100 µM CdCl_2_ for 24 h, Cd^2+^-induced NO production was investigated with the NO-sensitive fluorescent probe DAF-FM diacetate. Moreover, Cd^2+^-induced MAPK and caspase-3-like activities were inhibited in the presence of the NO-specific scavenger (cPTIO). These results prove that NO can quickly activate protein kinases with MAPK characteristics in *Arabidopsis* roots [33]. On the contrary, the caspase-3-like activity was significantly inhibited in *mpk6* mutants after Cd^2+^ treatment, and the tolerance of *Arabidopsis mpk6* mutants to Cd^2+^ and NO concentrations was also reduced [34]. 

In addition, this seriously affects the growth of rice seedlings and promotes the production of ROS and NO in rice roots after excessive Ar exposure. Subsequently, MAPK and MPK were activated in rice leaves and roots, respectively [35].

### 3.3. Plant Hormones

A number of phytohormones such as salicylic acid (SA), abscisic acid (ABA), auxin (IAA) and ethylene (ET) participated in important stress-related and developmental plant processes. The MAPKs homolog *AtMPK3* and *AtMPK6* of *Arabidopsis* are mainly involved in some environmental and hormonal responses [36]. As a homolog of *AtMPK6*, SIPK of tobacco has been proved to be a protein kinase induced by SA, which can be activated under environmental stresses [37]. Upon exposure to Cd, ABA could partially compensate the inhibitory effect of Cd on rice root growth, reduce auxin accumulation and affect the distribution of auxin. Moreover, the key genes of auxin signal transduction, including *YUCCA*, *PIN*, *ARF* and *IAA*, are negatively regulated by MAPK [38]. Moreover, ET and MAPK signal pathway–related genes were induced in soybean seedlings with Cd treatment. Subsequently, promoter sequence analysis showed that multiple regulatory motifs sensitive to ET and other plant hormones were found in *MAPKK2* [32]. 

## 4. Transcriptional Factors Regulate Heavy Metal Tolerance

MAPKs can phosphorylate various transcriptional factors in different abiotic stresses [44,45]. Transcriptional factors contain many phosphorylation sites and can regulate heavy metal stress by controlling the expression of downstream genes (Table 2). They also function as a central component in the regulatory networks of heavy metal detoxification and tolerance. Currently, many transcriptional factors with regard to heavy metal detoxification and tolerance have been found in plants. Among them, transcriptional factors such as basic leucine zipper (bZIP), heat shock transcription factor (HSF), WRKY, myeloblastosis protein (MYB) and ethylene-responsive transcription factor (ERF) have been known to play important roles in regulating heavy metal detoxification and tolerance in plants (Figure 1). 

### 4.1. bZIP

bZIPs, a large family of transcriptional factors in plants, are involved in a variety of biological processes and environmental challenges. A total of 135 bZIP-encoding genes were discovered by analyzing the whole genome and transcriptome of radish (*Raphanus Sativus*). Specifically, *RsbZIP010* exhibited downregulated expression under a variety of heavy metal stresses, such as Cd, Cr and lead (Pb) stresses [46]. In the *Glycyrrhiza uralensis* genome, 66 members of the *GubZIP* gene family were identified using a series of bioinformatics methods based on the hidden Markov model (HMM). Among them, 45 and 51 *GubZIP* genes were differentially expressed genes in roots and leaves under 0.02 g·kg^−1^ Cd stress, respectively [47]. 

Recently, *LOC_Os02g52780*, an ABA-dependent stress-related gene that belongs to the bZIP transcription factor family, was found by mapping QTL using 120 rice recombinant inbred lines and further testified to Cd accumulation in rice grains. The significant difference in the expression of the *LOC_Os02g52780* gene between parents indicates that they are related to the tolerance of rice to Cd stress, which may affect Cd accumulation in rice grains [48]. TGA (TGACG motif-binding factor) factor in *Arabidopsis* is a member of a subfamily of bZIP transcription regulators and is involved in the induction of pathogenic phase- and resistance-related genes [49]. When *Arabidopsis* responded to chromium (Cr^6+^) stress, bZIP transcription factor *TGA3* enhanced transcription of L-cysteine desulfhydrase (LCD) through a calcium (Ca^2+^)/calmodulin2 (CaM2)-mediated pathway and then promoted the generation of hydrogen sulfide (H_2_S), whereas H_2_S can trigger various defense responses and help reduce accumulation of HMs in plants [50]. In heavy metal accumulator *Brassica juncea*, the TGA3 homologous gene *BjCdR15* is upregulated in plants with Cd treatment for 6 h, indicating that *BjCdR15* transcription factor plays an irreplaceable role in regulating the absorption and long-distance transport of Cd. Western analysis showed that the abundance of *AtPCS1* protein increased significantly in Cd-treated plant shoots. Moreover, its overexpression confers tolerance and accumulation of Cd in *A. thaliana* and tobacco due to the regulation of the synthesis of phytochelatin synthase and the expression of several metal transporters [51]. *BnbZIP3* from *Boehmeria nivea* positively regulates heavy metal tolerance. On the contrary, *BnbZIP2* shows higher sensitivity to drought and heavy metal Cd stress during seed germination [52]. Additionally, bZIP transcription factor also interacts with other transcriptional factors (TFs) or mediates the downstream TFs to regulate Cd uptake. For example, bZIP transcription factor ABSCISIC ACID-INSENSITIVE5 (ABI5) interacts with *MYB49* and represses its function by preventing its binding to the downstream genes *bHLH38*, *bHLH101*, *HIPP22* and *HIPP44*, resulting in the inactivation of *IRT1* and a reduction in Cd uptake in *A. thaliana* [53]. In the zinc deficiency reaction in *A. thaliana*, two members of group F in the bZIP transcriptional factors, *bZIP19* and *bZIP23,* can bind zinc (Zn) ions to the zinc sensor motif and play the function of a central regulator [54,55].

### 4.2. MYB

MYB proteins are key regulators controlling development, metabolism and responses to biotic and abiotic stresses [83]. In rice, *OsMYB45* positively regulates Cd stress, and its mutant exhibited lower catalase (CAT) activity and higher concentrations of H_2_O_2_ in the leaves compared with the wild-type [56]. *SbMYB15,* from a succulent halophyte *Salicornia brachiata* Roxb, is an important heavy metal response gene. Overexpression of *SbMYB15* in transgenic tobacco can reduce the absorption of heavy metal ions Cd and nickel (Ni) and improve the scavenging activities of the antioxidative enzymes (CAT and SOD) [57]. *AtMYB4* regulates Cd tolerance by enhancing protection against oxidative damage and increases expression of *PCS1* and *MT1C* [58]. Furthermore, *JrMYB2* from *Juglans regia* is considered to be an upstream regulator of *JrVHAG1* that improves CdCl_2_ tolerance in plants. Under Cd treatment, the heterologous overexpression of *JrVHAG1* in *A. thaliana* showed a significant increase in fresh weight and primary root length and higher activities of SOD and POD compared with the wild-type [59].

As is a toxic metalloid in plants, usually in combination with sulfur and metals, and can be found in two inorganic forms, arsenite [As (III)] and arsenate [As (V)] [84]. Rice R2R3 MYB transcription factor *OsARM1* (ARSENITE-RESPONSIVE MYB1) regulates the absorption of As (III) and root-to-stem transport by regulating the As-associated transporters (*OsLsi1*, *OsLsi2* and *OsLsi6*) [61]. In *Arabidopsis*, *AtMYB40* negatively regulated the expression of *PHT1;1* (Pi transporter) and positively regulated the expression of *PCS1*, *ABCC1* and *ABCC2*, which acts as a central regulator conferring plant As (V) tolerance and reducing As (V) uptake [62].

In addition to Cd and As, MYB transcriptional factors are also involved in the homeostasis or absorption of essential elements such as Zn and iron (Fe). MYB72 is involved in metal homeostasis in *Arabidopsis*, and its knockout mutant was more sensitive to excess Zn or Fe deficiency compared to the wild-type [60]. Moreover, *DwMYB2* from the orchid can enhance Fe absorption as a regulator. In *DwMYB2*-overexpressing *Arabidopsis* plants, the Fe content in roots is two-fold higher compared to that in wild-type roots, while the reverse is true in shoots. This difference in Fe content between roots and shoots indicated that the translocation of iron from root to shoot in transgenic plants was regulated by *DwMYB2* [63].

### 4.3. WRKY

WRKY proteins, composed of a WRKY domain (WRKYGQK) and a zinc finger motif, can generally recognize the cis-acting W-box elements (TTGACC/T) of downstream genes. WRKY genes were found in many plant genome databases. A total of 126 WRKY genes have been found in the radish genome database. RT-qPCR analysis showed that 36 *RsWRKY* genes changed significantly under one or more heavy metal stresses. Specifically, 24 and 20 *RsWRKY* transcripts were induced under Cd and Pb treatments, respectively [64]. In soybean, 29 Cd-responsive WRKY genes were retrieved through the comprehensive transcriptome analysis of soybean under Cd stress. The overexpression of *GmWRKY142* in *A*. *thaliana* and soybean decreased Cd uptake and positively regulated Cd tolerance. Further analysis indicated *GmWRKY142* activated the transcription of *AtCDT1* (Digitaria ciliaris cadmium tolerance 1), *GmCDT1-1* and *GmCDT1-2* by directly binding to the W-box element in their promoters; however, *CDT1* rich in cysteine (Cys) proteins are important chelators of Cd [68]. Besides, *At**WRKY6* controls As (V) uptake through the regulation of Pi transporters while simultaneously restricting arsenate-induced transposon activation [70].

WRKY can enhance plant Cd tolerance or maintain the balance of metal ions by regulating downstream functional genes. *AtWRKY12* negatively regulates Cd tolerance in *Arabidopsis* though directly binding to the W-box of the promoter in *GSH1* and indirectly repressing phytochelatin synthesis–related gene expression [65]. Another WRKY transcription factor *AtWRKY13* enhances plant Cd tolerance by directly upregulating an ABC transporter *PDR8* [66] and promoting D-cysteine desulfhydrase and hydrogen sulfide production in *Arabidopsis* [67]. Additionally, *AtWRKY47* regulates genes responsible for cell wall modification (e.g., *XTH17*, *ELP*), which can maintain aluminum (Al) balance in ectoplasts and symplasts and improves Al tolerance [69].

### 4.4. HSF

Heat shock transcription factor is well known for responding to external heat stress. The member of class A has also been reported to be involved in the heavy metal stress response. A total of 22 Hsf members were identified in Cd/Zn/Pb hyperaccumulator *Sedum alfredii* and phylogenetically clustered into three classes, SaHsfA, SaHsfB and SaHsfC. In detail, 18 SaHsfs were responsive to Cd stress [71]. The expression levels of *SaHsfA4c* transcripts and proteins in all tissues were induced by Cd. Concurrently, it can upregulate ROS-related genes and HSPs, resulting in lower levels of ROS accumulation after Cd stress in transgenic *Arabidopsis* and non-hyperaccumulation ecotype *S. alfredii* [72]. *HsfA4a* in wheat and rice, all belonging to class A4a Hsfs, can confer Cd tolerance by upregulating metallothionine gene expression [73]. Transcriptome analysis of Cd-treated switchgrass roots showed that HSF/HSP was involved in the process of normal protein conformation reconstruction and intracellular homeostasis under Cd stress. Overexpression of an HSP gene in *Arabidopsis* significantly improved the tolerance of plants to Cd [74]. Transcription factor heat shock factor A1a (*HsfA1a*) can induce melatonin biosynthesis to some extent and endow tomato plants with Cd tolerance [75]. Moreover, *PuHSFA4a* from *Populus ussuriensi**s* regulates the target genes *PuGSTU17* and *PuPLA* to activate the antioxidant system and root development, thereby promoting excess-Zn tolerance in roots [76]. Therefore, the members of class HsfA enhance heavy metal tolerance by regulating the expression of key genes such as heavy metal chelators or antioxidants.

### 4.5. Other TFs

In addition to bZIP, MYB, WRKY and HSF, other transcription factor families also regulate the heavy metal response. In *Aegilops markgrafii*, overexpression of *AemNAC2* in wheat led to reduced Cd concentrations, thus contributing to Cd tolerance [77]. *Vigna umbellata* NAC-type TF, *VuNAR1*, confers Al resistance by regulating cell wall pectin metabolism [78]. Cd induces the expression of a C_2_H_2_ zinc-finger transcription factor, *ZAT6*, which could directly target GSH1 expression, thereby triggering Cd-activated PC synthesis in *Arabidopsis* [79]. Basic helix–loop–helix (bHLH) transcriptional factors *AtbHLH104*, *AtbHLH38* and *AtbHLH39* positively regulate genes involved in heavy metal absorption and detoxification [80,81]. There is also a complex regulation network between these transcriptional factors. For example, the ABA-mediated ABI5-MYB49-HIPP regulatory network repressed Cd uptake in *Arabidopsis* [76]. In *Phaseolus vulgaris*, ethylene responsive factors *PvERF15* and metal response element-binding transcription factor (MTF) *PvMTF-1* form a Cd-stress transcriptional pathway [82]. 

## 5. Conclusions

The MAPK cascade pathway is known to play an important role in plant growth, development and resistance to stress. For example, drought stress activates the MAPK cascade, phosphorylates selected targets and controls the activities of phospholipase, microtubule associated protein, cytoskeleton protein, kinase and other transcriptional factors in response to drought stress. A novel GhMAP3K15-GhMKK4-GhMPK6-GhWRKY59 phosphorylation loop that regulates the GhDREB2-mediated and ABA-independent drought responses in cotton has been identified [85,86]. However, in comparison with other abiotic stresses, there is little information about the MAPK cascade phosphorylating transcriptional factors in plant responses to heavy metals. Heavy metals have the characteristics of strong biological toxicity and rapid migration. They can lead to plant nutritional defects, inhibition of chlorophyll synthesis, reduction of photosynthesis, oxidative stress and, finally, inhibit plant growth and even result in death [87]. Plant roots sense heavy metal stress, trigger signal transduction and then cause a series of changes in physiological state and microstructure. Plant responses to HMs are regulated by the differential expression of genes, the enhancement of antioxidant-system activity and by the synergistic crosstalk between signal molecules. In depth understanding of the plants’ heavy metal stress perception, signal transduction and response processes are the prerequisites for plants to maintain stability under stress conditions [88]. The perception of heavy metal stress can trigger a variety of signal molecules in plants, such as NO, hormones, ROS, etc. These signaling molecules may activate the MAPK cascade. The kinase signal from upstream transmits to the downstream receptor and activates transcriptional factors, such as bZIP, HSF, MYB, WRKY, etc. These transcriptional factors promote the absorption, transport, isolation and detoxification of HMs by regulating downstream functional genes. These cascade responses involve complex and ordered mechanisms of the synergistic intracellular and extracellular regulation of homeostasis which is designed to translate extracellular stimuli into intracellular responses. Improving the chances of plant survival in heavy metal environments requires the activation of multiple defense responses. 

At present, the research on plant response to heavy metal stress has been carried out continuously. Although many signaling molecules are involved in plant responses to HM exposure, the exact nature of signal transduction is still unclear, as are the interactions between signal molecules and the functions of target proteins. In addition, there are still some gaps in our knowledge regarding the regulatory circuits of stress responses required for the protection of plant reproductive development. Therefore, it is necessary to explore a variety of ways to understand the tolerance and accumulation mechanisms of plants to HMs. In future studies, the key genes involved in HM accumulation should be further determined. At the molecular level, it is of great significance to clarify the interactions of signal transduction and signal cascades in plants with heavy metal exposure.

## Figures and Tables

**Figure 1 ijms-23-04463-f001:**
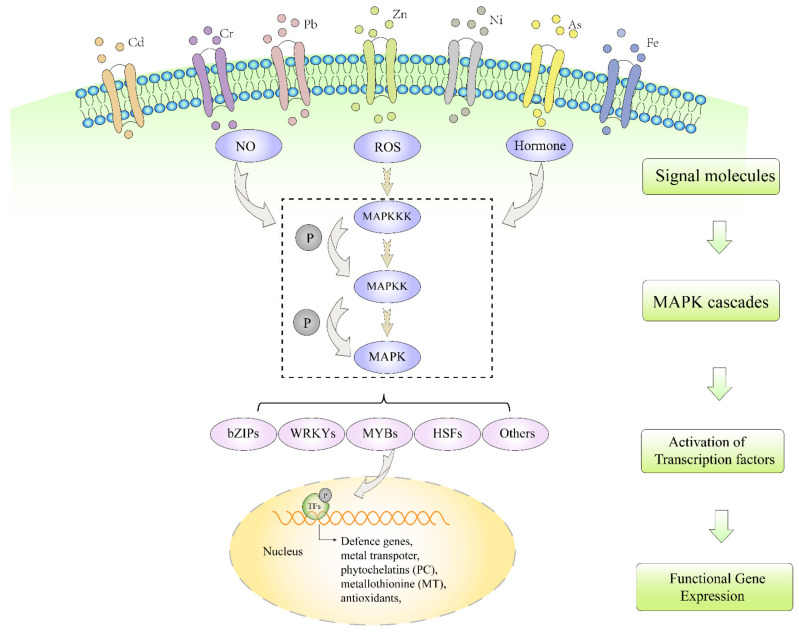
MAPK cascade and transcriptional factors in response to heavy metal stresses in plants. Heavy metal exposure triggers multiple signaling pathways such as NO, ROS and phytohormones. These signals interact and activate the MAPK cascade. Subsequently, MAPK cascade phosphorylates and activates related transcriptional factors including bZIP, WRKY, MYB, HSF and other transcriptional factors, which further induce the expression of defense genes, metal transporter genes, PCs, MTs, antioxidant related genes, etc. Finally, heavy metal tolerance or accumulation is enhanced in plants.

**Table 1 ijms-23-04463-t001:** Different signal molecules activate MAPK pathway under heavy metal stresses.

Signals	Heavy Metal	Plant	MAPK	Reference
ROS	-	*Arabidopsis thaliana*	*MEKK1-MKK4/5-MPK3/6*	[25]
	-	*Arabidopsis thaliana*	*MEKK1-MKK2-MPK4/6*	[26]
	Cu, Cd	*Medicago sativa*	*SIMK*, *MMK2*, *MMK3* and *SAMK*	[27]
	Cu	*Arabidopsis thaliana*	*MAPK*	[28]
	Cd	*Zea mays*	*ZmMPK3-1/ZmMPK6-1*	[29]
	Cd	*Arabidopsis thaliana*	*MPK3* and *MPK6*	[30]
NO	-	*Arabidopsis thaliana*	*AtWAKL10*	[31]
	-		*MAPK*	[32]
	Cd	*Arabidopsis thaliana*	*MAPK*	[33]
	Cd	*Arabidopsis thaliana*	*MPK6*	[34]
	Ar	*Oryza sativa*	*MAPK/MPK*	[35]
Hormone				
		*Arabidopsis thaliana*	*AtMPK3*/*AtMPK6*	[36]
SA	-	*Nicotiana tabacum*	*SIPK*	[37]
ABA/IAA	Cd	*Oryza sativa*	*MAPK*	[38]
ET	Cd	*Glycine max*	*MAPK/MAPKK2*	[32]

**Table 2 ijms-23-04463-t002:** Transcriptional factors in response to heavy metal stresses.

Family	Genes	Heavy Metals	Function	Number of Phosphorylation Sites	Reference
bZIP	*RsbZIP010*	Cd, Cr and Pb	*RsbZIP010* exhibited downregulated expression under Cd, Cr and Pb stresses.		[46]
	*GubZIP*	Cd	*GubZIPs* were expressed specifically in different tissues under cadmium stress		[47]
	*LOC_Os02g52780/OsbZIP23*	Cd	*LOC_Os02g52780* related to the tolerance of rice to Cd stress and affected Cd accumulation in rice grains.	39	[48]
	*BjCdR15*	Cr	*TGA3* elevates LCD expression and H_2_S production to bolster Cr^6+^ tolerance in *Arabidopsis.*	35	[49,50,51]
	*BnbZIP2 BnbZIP3*	Cd	Over expression of *BnbZIP2* exhibited more sensitivity to drought and heavy metal Cd stress.	0/44	[52]
	*ABI5*	Cd	*ABI5* interacts with *MYB49* and prevented its binding to the downstream genes, resulting in inactivation of *IRT1* and reduced Cd uptake.	46	[53]
	*bZIP19,23*	Zn	Zinc sensors to control plant zinc status.	25/17	[54,55]
MYB	*OsMYB45*	Cd	Under Cd stress, *OsMYB45* is highly expressed. Mutation of *OsMYB45* resulted in hypersensitivity to Cd treatment.	37	[56]
	*SbMYB15*	Cd, Ni	Overexpression of *SbMYB15* conferred. Cadmium and nickel tolerance in transgenic tobacco	45	[57]
	*AtMYB4*	Cd	*MYB4* regulates Cd-tolerance via the coordinated activity of improved anti-oxidant defense systems and through the enhanced expression of *PCS1* and *MT1C* under Cd-stress in *Arabidopsis.*	40	[58]
	*JrMYB2*	Cd	*JrMYB2* acts as an upstream regulator of *JrVHAG1* to improve CdCl_2_ stress tolerance stress tolerance.		[59]
	*AtMYB72*	Zn, Fe	The *Arabidopsis MYB72* knockout mutant was more sensitive to excess Zn or Fe deficiency than wild-type.	43	[60]
	*OsARM1*	As	*OsARM1* regulates arsenic absorption and root-to-shoot translocation.	19	[61]
	*AtMYB40*	As	*AtMYB40* enhances plant As (V) tolerance and reduces As(V) uptake.	28	[62]
	*DwMYB2*	Fe	The translocation of iron from root to shoot is affected by the *DwMYB2.*	39	[63]
WRKY	*RsWRKY*	Cd	*RsWRKY* transcripts were significantly elevated under Cd and Pb treatments.		[64]
	*AtWRKY12*	Cd	*WRKY12* represses GSH1 expression to negatively regulates cadmium tolerance in *Arabidopsis.*	31	[65]
	*AtWRKY13*	Cd	Activates *PDR8* expression to positively regulate cadmium tolerance in *Arabidopsis.*	49	[66]
	*AtWRKY13*	Cd	*WRKY13* activation of DCD during cadmium stress.	49	[67]
	*GmWRKY142*	Cd	*GmWRKY142* confers cadmium resistance by upregulating the cadmium tolerance 1-like genes.	54	[68]
	*AtWRKY47*	Al	A WRKY transcription factor confers aluminum tolerance via regulation of cell wall modifying genes.	63	[69]
	*AtWRKY6*	As	*WRKY6* transcription factor restricts arsenate uptake and transposon activation in *Arabidopsis.*	66	[70]
HSF	*SaHsfA4c*	Cd	The expression of *SaHsfA4c* was induced by cadmium and enhanced Cd tolerance by ROS -scavenger activities and shock proteins expression.	39	[71,72]
	*TaHsfA4a* *OsHsfA4a*	Cd	*HsfA4a* of wheat and rice confers Cd tolerance by upregulating MT gene expression.	46	[73]
	*PvBip1*	Cd	HSF/HSP participates in the reconstruction of protein conformation and improves intracellular homeostasis to increase cadmium tolerance.	64	[74]
	*HSF1A*	Cd	*HsfA1a* upregulates melatonin biosynthesis to confer cadmium tolerance in tomato plants.	55	[75]
	*PuHSFA4a*	Zn	*PuHSFA4* activates the antioxidant system and root development–related genes and directly targets *PuGSTU17* and *PuPLA.*	39	[76]
Others	*AemNAC2*	Cd	Overexpression of *AemNAC2* led to reduced cadmium concentration.	52	[77]
	*VuNAR1*	Al	*VuNAR1* regulates Al resistance by regulating cell wall pectin metabolism via directly binding to the promoter of *WAK1* and inducing its expression.	24	[78]
	*ZAT6*	Cd	*ZAT6* coordinately activates PC synthesis–related gene expression and directly targets GSH1 to positively regulate Cd accumulation and tolerance in *Arabidopsis*.	40	[79]
	*AtbHLH104* *AtbHLH38* *AtbHLH39*	Cd	*AtbHLHs* positively regulates genes involved in heavy metal absorption and detoxification.	27/27/35	[80,81]
	*HIPP22*	Cd	*MYB49* binds to the promoter regions of the *HIPP22* and *HIPP44*, resulting in upregulation Cd accumulation.	14	[76]
	*PvERF15*	Cd	*PvERF15* and *PvMTF-1* form a cadmium-stress transcriptional pathway.	44	[82]

## Data Availability

Not applicable.

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
