# Peer review of "MAPK Cascades and Transcriptional Factors: Regulation of Heavy Metal Tolerance in Plants"

_ijms, 2022, doi:10.3390/ijms23084463_

Round 1

Reviewer 1 Report

I think the study could give important information to the scientific community.

Please include some of the latest research findings and updated reviews during 2021-2022 needed in the introduction and discussion parts. For examples

https://scholar.google.com/scholar?as_ylo=2022&q=MAPK+Cascade+Heavy+Tolerance+in+Plants&hl=en&as_sdt=0,5

https://scholar.google.com/scholar?as_ylo=2021&q=MAPK+Cascade+Heavy+Tolerance+in+Plants&hl=en&as_sdt=0,5&as_rr=1

https://scholar.google.com/scholar?hl=en&as_sdt=0,5&as_ylo=2021&as_rr=1&q=MAPK+Cascade+Heavy+Tolerance+in+Plants

Name of all microbes should be italicized. Your manuscript must re-format based on MDPI style.

An English language revision is recommended since frequently the use of the grammar and the syntax results quite weird.

Furthermore, all figures must be updated to be error-free and of sufficient quality for publishing in microorganisms. Text inside figures must be readable.

Please provide a conclusion in the end of review and you must provide sufficient feedback on the main objectives of your study

Reviewer 2 Report

The reviewed paper deals with mitogen activated protein kinase. This problem has been studied and described many times. The authors presented in this paper a review of the literature on this problem. The paper is interesting and well written. However, I have questions:
1. How does the paper stand out from others in the same field.
2. Novelty please show in abstract and introduction.

Author Response

This manuscript is a resubmission of an earlier submission. The following is a list of the peer review reports and author responses from that submission.